# Validation of the Amharic version of perceived access to healthcare services for patients with cervical cancer in Ethiopia: A second-order confirmatory factor analysis

Tariku Shimels[1,2]*, Biruck Gashawbeza[3], Teferi Gedif Fenta[2]

1 Research Directorate, Saint Paul's Hospital Millennium Medical College, Addis Ababa, Ethiopia,
2 Department of Pharmaceutics & Social Pharmacy, School of Pharmacy, College of Health Sciences, Addis Ababa University, Addis Ababa, Ethiopia, 3 Department of Gynecology & Obstetrics, Saint Paul's Hospital Millennium Medical College, Addis Ababa, Ethiopia

* Tariku.shimels@sphmmc.edu.et

**Data Availability Statement:** All relevant data are within the manuscript and its Supporting Information files.

## Abstract

### Background

Accessing healthcare services is a multifaceted phenomenon involving various elements, encompassing the demand, identification, reach, and utilization of healthcare needs. The literature offers methods for capturing patients' perceptions of healthcare access. However, to accurately measure patient perceptions, it is imperative to ensure the validity and reliability of such instruments by designing and implementing localized language versions.

### Aim

The primary aim of this study was to validate the Amharic version of the perceived access to health-care services among patients diagnosed with cervical cancer in Ethiopia.

### Method

A cross-sectional study was conducted among cervical cancer patients at oncology centers in Addis Ababa, Ethiopia. A consecutive sampling approach was used and data collection took place from January 1 to March 30, 2023. Following initial validation and pretesting, a KoboCollect mobile phone application was employed for data collection. Subsequently, the collected data underwent cleaning in Microsoft Excel and analysis through Amos software v.26 and R programming. Various validity and reliability tests, such as content validity, convergent validity, face validity, divergent validity, known-group validity, and reliability tests, were executed. A second-order confirmatory factor analysis was developed to calculate incremental model fit indices, including CFI and TLI, along with absolute measures, namely SRMR and RMSEA.

### Results

A total of 308 participants were involved in the study, with 202 (65.6%) being patients referred from outside Addis Ababa. The initial evaluation of content validity by expert panels

**Funding:** Funding source for the present study was obtained from Addis Ababa University (AAU) and Saint Paul Institute for Reproductive Health and Rights (SPIRHR). The funders had no role in study design, data collection and analysis, decision to publish, or preparation of the manuscript.

**Competing interests:** The authors have declared that no competing interests exist.

indicated that all criteria were met, with a CVR range of 0.5 to 1, I-CVI values ranging from 0.75 to 1, an S-CVI value of 0.91, and face validity values ranging from 2.4 to 4.8. The internal consistency of items within the final constructs varied from 0.76 to 0.93. Convergent, known-group, and most divergent validity tests fell within acceptable fit ranges. Common incremental fit measures for CFI and TLI were achieved with corresponding values of 0.95 and 0.94, respectively. The absolute fit measures of SRMR and RMSEA were 0.04 and 0.07, indicating good and moderate fit, respectively.

## Conclusion

The study indicated a high internal consistency and validity of items with good fit to the data, suggesting potential accuracy of the domains. A five-domain structure was developed which enables adequate assessment of perceived access to health-care services of patients with cervical cancer in Ethiopia. We suggest that the tool can be utilized in other patient populations with a consideration of additional constructs, such as geographic accessibility.

## Background

Access to healthcare services is a multifaceted phenomenon encompassing broader concepts, involving the demand, identification, reach, and utilization of healthcare needs within the context of the ability to provide these services [1]. Frequently, access in the healthcare context has been viewed as the capacity of patients to reach services, healthcare providers, or institutions without encountering geographic, economic, or social barriers [2]. Gulliford et al. [3]. has argued that defining access in healthcare involves considering both vertical and horizontal equity factors across at least four dimensions: supply, utilization, acceptability, and affordability of services. Certain authors have presented clear and separate definitions of healthcare access. According to the theory by Penchansky and Thomas [4], access is achieved when the dimensions of affordability, availability, accessibility, accommodation, and acceptability, pertaining to both client and provider characteristics, are satisfied. Latter, an additional dimension, awareness, has been added by Saurman [5] as an integral component to access that should be considered along with the earlier dimensions.

Researchers have devised and suggested methods to capture patients' perceptions regarding access to healthcare services. Levesque et al. [1] introduced a more intricate assessment of access that took into account the capabilities of both recipients and providers. According to the authors, a comprehensive understanding of access involves measuring five dimensions of ability for accessibility (Approachability, Acceptability, Availability and accommodation, Affordability, Appropriateness) and the abilities of populations (Ability to perceive, Ability to seek, Ability to reach, Ability to pay, Ability to engage). On the other hand, a literature review by Quinn et al.(2017), highlighted that eleven main domains which six were related to access (access to primary care, specialty care, urgent care, spatial access, communication and cultural access) and five were related to care coordination(longitudinal, relational, informational, cross-boundary and follow-up) were reported [6]. An adaptation of previous theories has been utilized by Zandam et al. (2017) in Nigeria to develop a six-domain construct of access from users perspective [7]. In a more recent development, a Persian version of a questionnaire assessing perceived access to healthcare based on six domains was created, tested, and

validated in patients attending health centers [8]. A summary of the main definitions or dimensions proposed in the literature is presented in the table below (Table 1).

Given the descriptions and concepts provided thus far, individuals may perceive access to healthcare services differently in various contexts. Currently, perceptions of the primary domains, namely geographic accessibility, availability, affordability, acceptability, accommodation, and awareness, serve as crucial indicators influencing healthcare utilization, especially in Ethiopia. These domains are elucidated by various theories present in the literature, applicable in the developing context [9, 15]. Although there has been growing progress in various domains facilitating the measurement of access to healthcare resources, a gap exists in the validation of these tools for different contexts, where each component may be psychometrically understood inconsistently. The widespread adoption of validating a localized version of psychometric measures is particularly evident, especially in the case of cancer patients [16–18]. The approach ensures that the instrument is suitable, culturally sensitive, and reliable within the distinct context of the target population. Addressing this knowledge gap is a key objective of the study, especially in the Ethiopian context where the utilization of health services is considerably shaped by patients' perceptions [19, 20].

While ensuring the access of populations to health-care services still remains as a big challenge globally [21, 22], the problem faces a double-burden in Ethiopia augmented with

**Table 1. Summary of definitions or dimensions of access commonly reported in the literature.**

| Study ID. and reference | Summary of Definitions, Models, or Dimensions |
| --- | --- |
| Cabieses and Bird, 2014 [9] | The article outlined a glossary of terms and clarified the concept of access, underscoring its relevance to low and middle-income countries. |
| Buss et al., 2009 [10] | It delineates a managerial approach for healthcare access and emphasizes the administrative responsibilities essential for ensuring accessibility to health coverage, providing a detailed account of the execution of these tasks. |
| Gulliford et al., 2002 [3] | Highlighted the dependence of access on the affordability, physical accessibility, and acceptability of services. |
| Aday and Andersen, 1974 [11] | The article explored indicators related to access to healthcare services. In this framework, the dimensions of system and population were classified as process indicators, while utilization and satisfaction components were recognized as outcome indicators within a theoretical model of access. |
| Penchansky and Thomas,1981 [4] | Originally introduced, a definition stipulating that access is demarcated by five distinct dimensions: availability, accessibility, accommodation, affordability, and acceptability. |
| Levesque et al.,2013 [1] | Formulated the five dimensions of accessibility as follows: 1) Approachability; 2) Acceptability; 3) Availability and accommodation; 4) Affordability; 5) Appropriateness. Simultaneously, the Levesque framework established five corresponding abilities of populations to facilitate access: 1) Ability to perceive; 2) Ability to seek; 3) Ability to reach; 4) Ability to pay; and 5) Ability to engage. |
| Beth E Quill, 2008 [12] | Defined access to healthcare as the achievement of individual utilization of healthcare services and the factors that either enable or impede one's health. |
| IOM,1993 [13] | Outlined access to medical care as the timely utilization of individual health services aimed at achieving optimal health outcomes. Emphasized that such access should be marked by safety, effectiveness, timeliness, patient-centeredness, efficiency, and equity. |
| Anderson, 1968 [14] | Proposed the initial three-stage behavioral model of healthcare access, utilizing predisposing, enabling, and need components to explain the varied utilization of medical care services by families. |
| Saurman, 2016 [5] | Modified Penchansky's access model by integrating the component of awareness. |
| Quinn et al., 2017 [6] | Expressed that metrics like 1) acceptability of the distance to the care site, 2) the burden on patients to access and coordinate care and billing, and 3) provider familiarity with veteran culture and processes are limited in standard assessment instruments. |

inequitable accessibility of inputs [23, 24] and poor uptake of services on the other end [25]. This may be attributed to differences in the level of health facilities, economic status of patients, provider-related issues, and causes rooted in local contexts. While the healthcare system in Ethiopia primarily emphasizes a preventive approach, initiatives to enhance tertiary care services, especially for individuals with cancer, are in their early stages. The country introduced its initial national cancer control plan, aligning with the national priorities of the Health Sector Transformation Plan (HSTP) 2015/16-2019/20. The goal was to reduce the incidence and mortality of cancer through comprehensive preventive and treatment strategies [26]. As yet, the incidence of cervical cancer and its associated risk factors is increasing, establishing it as the second most prevalent and the second most fatal cancer in the country according to a 2018 report [27]. A recent qualitative study highlighted that lack of awareness, misconceptions, and negative perceptions were common reasons for the low uptake of cervical cancer screening [20]. Another study reported that patients had low perceptions of risk to cervical cancer while availability of equipment, space and training for providers [28]. Due to the abstract and comprehensive nature of the concept, the existing reports in Ethiopia may not provide a full picture of the issue under consideration. This could be due to reports focusing on only certain domains, employing solely qualitative methodologies, or using non-validated tools. Consequently, comprehending the extent of the problem and its contributing factors may pose a challenge, given the lack of standardized and validated tools, especially for patient reported outcomes that involve a psychometric component. Psychometric measures facilitate the modeling and establishment of how psychological constructs, like human intelligence or psychological abilities, can be quantitatively estimated based on a set of observable scores [29]. Their function in illustrating individual cognitive variations and attitudes toward a behavior can assist researchers in making informed decisions when crafting a study, including determining the optimal number of trials for a task [30]. Additionally, measuring patients' perceptions of the care provided at tertiary centers is highly significant for identifying their expectations and designing appropriate interventions [31, 32].

The objective of this study was to validate the psychometric assessment of the Amharic-version regarding patients' perceived access to healthcare services in the Ethiopian context. Access to healthcare services was treated as a second-order latent factor, incorporating additional latent domains reported in the literature as first-order factors.

## Methods

### Study context, design, and population

This research was conducted at the oncology centers of two tertiary public hospitals, Tikur Anbessa Specialized Hospital (TASH) and Saint Paul's Hospital Millennium Medical College (SPHMMC) in Addis Ababa, Ethiopia. These hospitals are among the top referral units in the country, providing specialized and comprehensive healthcare services along with medical training. Established in 1969 by the German Evangelicals, SPHMMC is currently under the administration of the federal Ministry of Health of Ethiopia (FMOH). In addition to its academic functions, it offers a diverse range of healthcare services with 392 beds, serving an annual average of 200,000 patients and a catchment population of over 5 million [33]. The hospital, currently, provides all chemotherapy services except radiotherapy to patients diagnosed with cervical cancer. TASH is likely the oldest and largest referral hospital in the country, boasting 700 beds and being owned by Addis Ababa University. In addition to providing training across various health disciplines, the hospital offers healthcare services to patients referred from all parts of the country. The hospital's dedicated oncology unit, situated in the 5[th] police station, provides both inpatient and outpatient oncology services, covering radiotherapy,

chemotherapy, surgery, and chemoradiotherapy [34]. A cross-sectional design was implemented between 1st January and 30th March 2023 to validate the 30-item Amharic version of perceived access to health-care services in patients with cervical cancer. Efficient and swift, this design was deemed suitable for our study, aiming to gauge the validity of constructs related to access to healthcare services. The assessment occurs at a specific point in time, capturing a snapshot of the data from patients. All patients with a confirmed diagnosis of cervical carcinoma, undergoing follow-up during the study period, and consenting to participate, were consecutively enrolled. Although not a strictly random selection technique, consecutive sampling was chosen in this study because the participants shared homogeneity in terms of diagnosis, the patient population's number and flow were limited and stable, ensuring temporal representation during the three-month data collection period. This technique also reduces selection bias, provided that the primary referral settings were simultaneously included. As studies suggested that a participant to item ratio of 5:1 or that a sample size of at least 200 would be adequate for factorial validity [35, 36], a total sample of 308 patients was recruited and assessed.

## Validity and reliability of the instrument

Ensuring the validity and reliability of any quantitative research tool is a crucial step to assure the internal and external validity of findings. This is especially pertinent in cases like ours, where the tool predominantly incorporates a hierarchical and abstract psychometric component. This, in turn, plays a significant role in generating robust and accurate results that can be replicated [37]. The validity and reliability of the instrument used in data collection have been emphasized, employing a variety of approaches sequentially to ensure its accuracy and consistency. Presently, the concept of access to healthcare remains intricate, and the proposed domains and items may not comprehensively capture its complexities. Various additional factors may still influence, potentially resulting in insufficient scoring or factor loading. In some cases, an item might show reliability without indicating validity, posing a challenge when trying to generalize results. Despite this, the steps highlighted in the guidelines by Tsang et al. [38], and Yusoff and colleagues [36] were adhered to during the adaptation and validation process of the questionnaire. In general, the entire process of instrument development has undergone a sequence of five key steps, which include instrument design and adaptation, translation, validity testing of the draft instrument, piloting, and psychometric evaluation of the construct. The specific method, whether qualitative or quantitative, to be employed, along with the detailed steps and necessary inputs during each stage, are described below.

## Instrument design and adaptation

The psychometric domains and constructs for assessing patients' perceived access to healthcare services have been adapted from a recently developed and validated questionnaire by Hoseini and colleagues [8]. The tool has demonstrated satisfactory fit in the Persian context, where the health system and economic conditions are quite similar to ours. In fact, some sections of the tool were designed and recommended by previous researchers for assessing access to healthcare [1, 4]. The earlier instrument, which focused on primary healthcare services, has been modified to suit the current study settings, particularly secondary healthcare services for patients with cancer. Consequently, the constructs were adjusted to be applicable to any health facility, not restricted to health centers. The items assessing availability were entirely revised to inquire about the availability of medicines, an adequate number of staff, and the skills of health personnel at the settings. The first domain, originally named 'accessibility,' has been changed to 'reachability' to better reflect geographic access to services. Additionally, the first item, 'the services I request are given at the health center,' was removed as it was considered redundant

with specific inquiries under the 'availability' domain. Furthermore, an item in the 'acceptability' domain has been revised to account for religious diversity in the current setting, in addition to the cultural differences mentioned in the earlier version. As details on the contents of sections 1 and 2 of the instrument have been provided in previous sections, the focus of validation and subsequent descriptions will be on the adapted psychometric instrument. To navigate cultural nuances during the translation and validation processes, various measures were implemented. These included engaging experts in public health and research with a background in tool translation, consulting with senior experts familiar with the local health system, ensuring that all experts involved in the tool content validation were informed about the study objectives, incorporating back-translation, and conducting pilot testing of the instrument.

### Forward translation to Amharic version

Two translators took part in the translation process of the initially adapted psychometric instrument into Amharic. In an effort to minimize disparities in content interpretation, the principal investigator (PI), possessing a master's degree in pharmacoepidemiology and considerable experience in translating research instruments, undertook the initial translation of the original content. Subsequently, a public health specialist, well-versed in tool translation and qualitative research, and without prior knowledge of the study's objective, conducted the second translation. Any discrepancies arising between the two translations were addressed through discussion or by seeking the input of a third translator.

### Backward translation

In this phase, the Amharic version of the instrument underwent back-translation into English to ensure the consistency of items with the adapted version and to identify any unforeseen deviations. Two public health specialists, well-versed in research methodology and unfamiliar with the study objectives and the intended message of the entire instrument, participated in the back-translation process.

### Content validity

Validity, by definition, refers to the extent to which collected data accurately represents the phenomenon under investigation or "measures what is intended to be measured" [39]. Both qualitative and quantitative methods of content validity have been implemented, as they were employed by a panel of experts in the primary psychometric evaluation of the original tool [8]. The purpose of this assessment was to evaluate the instrument, in a multidisciplinary approach, for potential use in the Ethiopian context. It is recommended that the content validation process includes four essential steps: domain definition, domain representation, domain relevance, and appropriateness of the test construction procedure [40]. As a qualitative method for ensuring validity, a purposively selected group of health professionals, including physicians, nurses, pharmacists, researchers, and public health experts, were invited to assess the domain content, wording, appropriate placement of items, and relevance for measuring perceived access to healthcare services, specifically in patients with cervical cancer. The selection of the expert panel was specifically guided by the contents of the tool from the perspectives of the mentioned professions, some of whom had experience in cancer care and research. Subsequently, individual information was collected and incorporated into the tool before proceeding to quantitative evaluation. The feedback received from the panel played a crucial role in refining and enhancing the final instrument. A systematic process was followed to incorporate the feedback, which included thorough reviews of each input, prioritization based on relevance and potential impact, discussions with panel experts, iterative revisions, and validation.

The quantitative process of content validation in this study has undergone the necessary steps outlined elsewhere, including domain definition and representation. The approach was carried out using both a face-to-face (by panel of experts) and a non-face-to face approach (using an electronic assessment form) [36] and was centered on assessing the essence, relevance, and appropriateness of the scale.

The evaluation criterion for item relevance was structured using a 3-level Likert scale: "item is essential," "item is useful but not essential," and "item is not essential." Subsequently, content validity ratio (CVR) and content validity index (CVI) values were calculated. The formula for content validity ratio was CVR = (Ne—N/2) / (N/2), where Ne is the number of panelists indicating that the item is "essential," and N is the total number of panelists, as documented earlier by Lawshe [41]. The decision of whether to include an item in the scale depended on the number of experts and the corresponding minimum ratio recommended in Lawshe's table. Panelists or experts were furnished with an evaluation format to assess the level of agreement they could report based on the essence of the items in question.

Similarly, we employed a widely recognized method of content validity known as the content validity index (CVI). The CVI assesses the relevance and clarity of items. According to Davis [42], a 4-point ordinal scale (1 [not relevant], 2 [somewhat relevant], 3 [quite relevant], and 4 [highly relevant]) was incorporated to evaluate both dimensions.

Utilizing the scale provided in the table, two measures were computed: the item-level content validity index (I-CVI) and the scale-level content validity index (S-CVI). The I-CVI indicates the proportion of agreement on the relevance of each item, ranging between 0 and 1, while the S-CVI is defined as "the proportion of total items judged content valid." An I-CVI of 80 percent or higher is generally considered "appropriate" [43]. Some studies suggest that 78% (0.78) is an acceptable lower threshold for I-CVI, while S-CVI is considered excellent when it achieves a score of at least 80% [44].

## Face-validity

Face validity was assessed to determine if the instrument ultimately gains acceptability based on its face value by the respondents. This aspect of validity addresses whether an instrument appears valid to subjects, patients, and/or other participants [38]. One can address this question from the participants' perspective by assessing their agreement with the items and their wording in the instrument to fulfill the research objectives. Another definition of face validity is the extent to which a test seems to measure what it asserts to measure. It provides a broad evaluation as a quick assessment of what the test aims to measure [45]. Face validity can be assessed using a 5-point Likert questionnaire with options such as 'very important,' 'important,' 'relatively important,' 'slightly important,' and 'unimportant.' An item impact score is calculated by multiplying the frequency and importance. Frequency is determined by the percentage of participants with a score of 4 or 5, while importance is assessed by averaging the sum of scores for each item. Ultimately, the item impact score is obtained by multiplying these two results. The general guideline is that if the item impact of an item is equal to or greater than 1.5 (mean frequency of 50% and an importance mean of 3 on the 5-point Likert scale), it is advisable to retain it in the instrument; otherwise, it may be eliminated [46].

## Piloting the instrument

A sample size of 30 participants was employed in this test, a size deemed suitable for identifying any issues (such as ambiguities or misunderstandings) in the instrument, as per documented acceptability [47]. In this phase, the demographic makeup closely mirrored that of the broader study population with respect to variables like diagnosis, study setting, and source

population. However, it's crucial to emphasize that the pilot sample was separate and not included as part of the official study population. The primary objective of the pilot testing was to verify the face validity and internal consistency of the items and assess the duration required for an interviewer to complete the questionnaire. According to Yusoff [36], acquiring a standardized method for distributing the tool and conducting interviews, particularly for interviewer-administered questionnaires, was emphasized. Additionally, improvements were made to the flow, process of items, administration, and logistical aspects of the fieldwork during this phase. Accordingly, altering the sequence some items, determining the estimated time that an interview would take, and the management of the data collection process through the Kobo-Collect application were among the areas revised during this time. The rationales behind the modifications were to ensure logical flow of items, improve responders' trust, and ensuring the quality of the data collected. Finally, at this phase, a preliminary internal consistency test was performed to make sure if items are eligible for further development. This was done so because Cronbach's alpha is not an absolute measure of consistency under all circumstances [48], and variability across different samples is always inevitable-hence the need to test reliability of the measure for both the pilot and the construct validation. While the use of Cronbach's alpha was meant to check and improve the internal consistency of items per each domain, the confirmatory factor analysis was to assess the underlying construct validity of the instrument. In the context of our study, the goal was to measure specific constructs related to access to healthcare, and the CFA allows us to confirm that the observed variables (survey items) align with the hypothesized latent constructs, both at the first and the second order levels.

## Initial reliability analysis of factor constructs

Reliability assesses the internal consistency of items within a scale, indicating whether the items in the instrument are conceptually compatible. In this study, the internal consistency of item constructs was evaluated using Cronbach's alpha. While a sample size of 20 to 30 patients is considered sufficient for evaluating the reliability of a survey instrument, as recommended elsewhere [49], we used the entire sample of respondents recruited as part of the factor analysis above, which also enhances the precision of estimates. In doing so, an alpha value of at least 0.7 was deemed acceptable [50].

## Confirmatory factor analysis

Construct validity is a crucial concept in assessing a questionnaire designed to measure an unobservable construct, often referred to as latent variables. Evaluating the construct validity of a new instrument involves estimating its association with other measures within the same domain, typically utilizing a different scale. This correlation may manifest as positive, negative, or null [48]. As such, this study assessed four types of construct validity: convergent validity (positive correlation), divergent (discriminant) validity (poor correlation), known-group validity, and factorial validity.

Derived from content analysis, factorial validity was utilized in this study as the suitable method for evaluating the validity of item constructs within the domains of access to healthcare services. This choice was made due to the presence of 30 items distributed across six latent domains associated with the concept of perceived access to healthcare. In the analysis of factorial validity, it is expected that several items intended to measure a specific dimension within a construct of interest would exhibit higher interrelatedness compared to those measuring other dimensions [7, 48]. This implies that factor analysis unveils relationships among items, between items and constructs, and among different constructs [51].

The interpretation of results from the confirmatory factor analysis (CFA) was as follows: Model fit was considered good if the Tucker Lewis Index (TLI), Normed-Fit Index (NFI), non-normed-fit index (NNFI), and goodness of fit (GFI) were ≥ 0.95 or acceptable if ≥ 0.90. Comparative fit index (CFI) and adjusted goodness of fit (AGFI) were considered acceptable if ≥ 0.90. For the root mean square error of approximation (RMSEA), a fit was considered good if ≤ 0.05 and moderately acceptable if between 0.05 and 0.08. The ratio of $\chi 2$ to degrees of freedom ($\chi 2/df$) was deemed the best fit when below 2 and moderate fit when it ranged between 2 and 3 [52, 53]. To distinguish the discriminant validity across included latent variables in the model, Henseler's later heterotrait-monotrait (HTMT2) ratio method [54] was utilized.

## Data collection process and quality assurance

After translation and initial content and face validation, data for the study were gathered using an interviewer-administered structured instrument created in the KoboCollect mobile-based application. Subsequently, the data were managed through the KoboToolbox server [34]. Five data collectors, consisting of three individuals with master's degrees and two general practitioners, were engaged in the data collection process. A one-day training session was conducted for both data collectors and supervisors, encompassing a comprehensive orientation to research ethics, study objectives, and familiarity with the instrument. This thorough training aims to equip data collectors and supervisors with the essential skills, understanding, and cultural awareness needed for consistent and accurate data collection, thereby enhancing the overall quality and reliability of the study findings. In addition to pretesting and training, continuous supervision of the data collection process and adherence to the study protocol were implemented to ensure quality.

## Data analysis

Data was cleaned and coded in MS-Excel for windows, R programming [55], and Amos software v.26 for analysis [56]. Descriptive statistics were utilized to illustrate participants' profiles, while structural equation modeling (SEM) was employed to present model fitness indices, a standardized correlation matrix, and the structure of the construct. Given the violation of normality assumptions, measurements were compared against bootstrapped estimates with confidence intervals and standard errors of the sample. Known group validity was confirmed using Student's t-test (for two-grouped variables) and one-way analysis of variance (one-way ANOVA) for variables with more than two categories. The estimates in the t-test were based on the second row (assuming unequal variances), while the post-hoc pairwise comparison in ANOVA utilized Dunnett's T3 calculation [57]. The computation of divergent validity was estimated the methods proposed by Henseler et al. [54]. The application of the mentioned models and specified analysis methods was considered suitable for the nature of the collected data, aiming to generate quantitative findings relevant to the research question objectives.

## Ethics consideration

The study received ethics approval from Saint Paul's Hospital Millennium Medical College (SPHMMC) (Ref.no: PM23/284), and a support letter was issued to Tikur Anbessa Specialized Hospital (TASH) (Ref.no: PM23/295) as part of the mini-grant offer obtained from Saint Paul Institute for Reproductive Health and Rights (SPIRHR). Verbal informed consent was obtained from all patients participating in this study. The informed consent process entails a comprehensive explanation of the study objectives, procedures, and potential risks and benefits to participants. Verbal consent requests are presented to participants, and any questions

they may have are addressed before securing their voluntary agreement to participate. To uphold confidentiality and privacy, participant identities are anonymized, data is securely stored, and access is restricted to authorized personnel. These measures are implemented to safeguard sensitive information and maintain ethical standards throughout the study.

## Results

### Characteristics of participants

A total of 308 participants took part in the current survey. Among them, 202 (65.6%) were referred patients from outside of Addis Ababa. The mean (SD) age of the patients was 50 (±11) years, ranging from 24 to 83. Over half (57.8%) were married, and approximately half (51%) had basic literacy skills. Additionally, more than two-thirds (69.5%) reported having a family size larger than 5. The median monthly income for households was reported to be 4000 ETB, with a mean (SD) of 4539 (±3325) ETB. About 55.5% of the respondents surpassed the extreme poverty line according to the World Bank's income classification for low-income countries. Among the respondents, 138 (44.8%) were housewives, and nearly three-quarters (74%) had coverage under community-based health insurance (CBHI). Examining clinical profiles, 128 (41.6%) were in stage II of the disease, followed by 111 (36%) in stage III. The majority (64%) had no history of any chronic comorbidity (see Table 2). In broad terms, the demographic profile of participants indicates that a significant portion of oncology center attendees were referred from various regions of the country, were married, had family sizes exceeding 5, earned monthly incomes above the extreme poverty line, possessed coverage under CBHI, and had a history of comorbidities. Consequently, the validated instrument is expected to be applicable and relevant to patients with similar characteristics throughout the country.

### Initial validity and reliability assessment

In this phase, a series of qualitative and quantitative validity assessments were conducted. Seven experts with diverse specializations, including clinical (MD), public health research (MPH), qualitative methods (MPH), pharmacy service (BPharm), Microbiology (MSc), pharmacoepidemiology (MSc), and lab quality assurance (MSc), were invited to evaluate the content and face validity of the 30-item adapted instrument. Two experts were involved in both forward and backward translation processes. After modifying two components (changing the factor name from 'access' to 'reachability' or referring to geographic access and altering 'screening for cervical cancer' to 'breast or colon cancer'), quantitative measures were derived from expert opinions. The first adjustment aimed to prevent confusion. In the earlier version, the term 'access' referred specifically to geographic access, causing confusion with the broader concept of access covering all constructs under investigation. In the second modification, we opted to replace "screening for patient breast or colon cancers" instead of cervix. This change was made to align with the study population, which consists of patients already diagnosed with cervical cancer. The Content Validity Ratio (CVR) calculated ranged from 0.5 to 1, while the Item-Level Content Validity Index (I-CVI) values varied from 0.75 to 1. The Scale-Level Content Validity Ratio (S-CVR) was 0.91. Additionally, face validity scores ranged from 2.4 to 4.8. The obtained content validity ratio (CVR) and item-level content validity index (I-CVI) values suggest strong content validity for the instrument. Higher values indicate a consensus among experts on the necessity and relevance of the items, affirming the quality of the instrument in assessing the intended constructs related to healthcare access in the study context. Following content and face validity assessments, the instrument was distributed to a sample of 30 participants to measure its reliability. The Cronbach's alpha for the entire scale was 0.72. Higher

**Table 2. Characteristics of study participants among patients visiting oncology centers in Addis Ababa, Ethiopia.**

| Characteristic | Label | Frequency | Percent |
|---|---|---|---|
| Residence | | | |
| | Addis Ababa | 106 | 34.4 |
| | Outside Addis Ababa | 202 | 65.6 |
| Age category (Yrs.) | | | |
| | 50 or less | 184 | 59.7 |
| | >50 | 124 | 40.3 |
| Marital status | | | |
| | Married | 178 | 57.8 |
| | Single | 15 | 4.9 |
| | Separated | 10 | 3.2 |
| | Divorced | 33 | 10.7 |
| | Widowed | 72 | 23.4 |
| Literacy status | | | |
| | Yes | 157 | 51.0 |
| | No | 151 | 49.0 |
| Number of family members | | | |
| | 5 or less | 214 | 69.5 |
| | > 5 | 94 | 30.5 |
| Average monthly income* | | | |
| | 3567 or less ETB | 137 | 44.5 |
| | >3567 ETB | 171 | 55.5 |
| Presence of social support | | | |
| | Yes | 167 | 54.2 |
| | No | 141 | 45.8 |
| Employment status | | | |
| | Government employee | 18 | 5.8 |
| | Private employee | 53 | 17.2 |
| | House wife | 138 | 44.8 |
| | Jobless | 49 | 15.9 |
| | Others** | 50 | 16.2 |
| Current status of CBHI coverage | | | |
| | Yes | 228 | 74.0 |
| | No | 80 | 26.0 |
| Stage of cancer | | | |
| | Stage I | 48 | 15.6 |
| | Stage II | 128 | 41.6 |
| | Stage III | 111 | 36.0 |
| | Stage IV | 21 | 6.8 |
| Presence of any chronic comorbidity | | | |
| | Yes | 114 | 37.0 |
| | No | 194 | 63.0 |

*classification was based on the 2022 update on extreme poverty classification (USD 2.15/person/day) for low income countries [58] and the USD and Birr exchange rate in August 2023.

** indicates merchants or farmers.

**Table 3. Factor-level reliability test assessment of indicators on perceived access to healthcare among patients visiting oncology centers in Addis Ababa, Ethiopia.**

| Factor | Number of items | Factor-level reliability (std. alpha) | Item label | Reliability if an item is dropped (std. alpha) | Remark |
|---|---|---|---|---|---|
| Availability | 2 | 0.76 | AV3 | 0.62 | Retained after removal of AV1 and AV2 |
| | | | AV4 | 0.62 | |
| Acceptability | 7 | 0.93 | AC1 | 0.92 | Retained after removal of items AV6 and AV7 |
| | | | AC2 | 0.92 | |
| | | | AC3 | 0.91 | |
| | | | AC4 | 0.92 | |
| | | | AC5 | 0.92 | |
| | | | AC8 | 0.92 | |
| | | | AC9 | 0.92 | |
| Affordability | 3 | 0.87 | AF1 | 0.75 | No item has been dropped |
| | | | AF2 | 0.73 | |
| | | | AF3 | 0.95 | |
| Accommodation | 3 | 0.76 | ACC1 | 0.81 | Retained after removal of items ACC2, ACC3, and ACC6 |
| | | | ACC4 | 0.63 | |
| | | | ACC5 | 0.56 | |
| Awareness | 4 | 0.90 | AW1 | 0.86 | Retained after removal of AW5 |
| | | | AW2 | 0.91 | |
| | | | AW3 | 0.85 | |
| | | | AW4 | 0.87 | |

alpha values, typically greater than 0.7, reflect stronger internal consistency, reinforcing the instrument's reliability in consistently measuring the factors under investigation.

## Internal consistency of factors

As illustrated in Table 3 below, an assessment of factor-level reliability was conducted to evaluate the internal consistency of item constructs within each of the six domains. If the removal of an item resulted in an improvement in factor-level consistency, that specific item was excluded. However, all factors were required to exhibit a Cronbach's alpha of at least 0.7 to qualify for subsequent factor analysis. Additionally, a factor loading of at least 50% in the structure matrix was expected. Despite the factor 'reachability' achieving an acceptable Cronbach's alpha value of 0.98 with two items, it was excluded from the final second-order structure matrix due to its very low loading (0.12).

## Convergent validity

The assessment of convergence among the items within each factor and the latent variables explaining perceived access to healthcare services involved evaluating composite reliability (CR) and average variance extracted (AVE). Upon examination of these measures, it was observed that all factors met the recommended minimum threshold of 0.6 for CR, signifying a high level of reliability for the scale. This measure aligns with Cronbach's alpha values, suggesting good construct reliability. Similarly, the AVE for each scale was computed, and the findings indicated that the average reliability of each factor exceeded the recommended threshold of 0.5, affirming model fit. In conclusion, both CR and AVE measures support the notion that the construct fits the data well. Additionally, the higher CR compared to AVE suggests strong reliability. The latent variables of the first-order scale were treated as inputs for the second-order model (Table 4).

**Table 4. Convergent validity of latent factors in perceived access to health-care services in patients with cervical cancer visiting oncology centers in Addis Ababa, Ethiopia.**

| Factor | Composite reliability (CR) | Average variance extracted (AVE) |
|---|---|---|
| Availability | 0.67 | 0.67 |
| Acceptability | 0.93 | 0.67 |
| Affordability | 0.88 | 0.72 |
| Accommodation | 0.77 | 0.54 |
| Awareness | 0.90 | 0.69 |
| Perceived access to health care | 0.95 | 0.80 |

## Divergent validity

The instrument's divergent validity was assessed using the updated criterion of the hetero-trait to mono-trait (HTMT2) ratio, as proposed by Henseler et al. (48). This method involves calculating the ratio of the sum of intra-factor level correlations to inter-factor level correlations to gauge the discriminant ability of a latent variable in comparison to other latent variables. Among the ten pairs of factors examined, eight demonstrated an acceptable level of discriminant validity (below 0.90), while two pairs, specifically acceptability vs. awareness and acceptability vs. availability, exhibited a slightly higher value (0.91) compared to the recommended threshold mentioned earlier (Table 5). While this ratio helps determine the extent to which different latent variables discriminate from each other, the implications of the two factor pairs, acceptability vs. awareness and acceptability vs. availability, showing higher values (0.91), suggest a potential overlap or shared variance between these constructs. This may indicate some degree of interrelatedness or conceptual similarity between acceptability and awareness, as well as acceptability and availability.

## Known-group validity

The instrument's validity was assessed by examining its ability to discriminate between various groups within the same sample, anticipating differences in their perceptions of access to healthcare services. These groups included income category, health insurance coverage, social support presence, and occupation. The construct effectively differentiated between groups based on average monthly income, with those above the extreme poverty line scoring significantly higher than those below the line (t = -5.95; df = 187.8; p<0.001). Similarly, individuals with current community-based health insurance (CBHI) coverage exhibited a relatively low mean aggregate score, indicating a poorer perception of access to healthcare services (t = -5.65 df = 200.7; p<0.001). Moreover, respondents reporting any form of social support had a significantly higher mean aggregate score for perceived access to healthcare services (t = 3.39; df = 186.4; p = 0.001). Furthermore, a post-hoc analysis using one-way ANOVA with

**Table 5. Divergent validity of latent factors in perceived access to health-care services in patients with cervical cancer visiting oncology centers in Addis Ababa, Ethiopia.**

| Awareness | | | | |
|---|---|---|---|---|
| Accommodation | 0.76 | | | |
| Acceptability | 0.91 | 0.81 | | |
| Affordability | 0.74 | 0.66 | 0.79 | |
| Availability | 0.85 | 0.76 | 0.91 | 0.74 |
| | Awareness | Accommodation | Acceptability | Affordability | Availability |

Dunnett's T3 comparison method revealed variations in perceived access to healthcare services among different occupation groups. Specifically, government employees had a significantly higher mean aggregate score compared to housewives (mean difference = 7.10; 95% CI = 1.85–12.27; p = 0.003) or those with no job at all (mean difference = 12.72; 95% CI = 5.16–20.18; p<0.001). The findings from the known group validity analysis indicate that the instrument effectively distinguishes between groups based on income, health insurance status, social support, and occupation. Higher mean scores among individuals with income above the extreme poverty line, those without current community-based health insurance (CBHI) coverage, and those reporting any form of social support suggest a more favorable perception of access to healthcare services in these groups. Additionally, the instrument highlights significant variations in perceived access to healthcare services among different occupation groups, with government employees scoring notably higher compared to housewives or those without employment. Overall, these results demonstrate the instrument's ability to discriminate between diverse groups based on their characteristics and perceptions of healthcare access.

## Model fitness indices

The overall fit of the construct was evaluated using various parameters, as outlined in Table 6. Despite the chi-square statistics for goodness of fit not rejecting the null hypothesis of good fit (pCMIN<0.001), the ratio of minimum discrepancy to the degree of freedom fell within the range of 2<CMIN/df<3, indicating a well-fitting model. The normed fit index (NFI), a size-insensitive measure of fitness, yielded a favorable value of 0.93, suggesting that the present model was a good fit. An alternative adjustment to the NFI using the incremental fit index (IFI) and comparative fit index (CFI), both considering sample size and degrees of freedom, resulted in higher values (0.95), indicating a well-fitted model. Additionally, the Tucker-Lewis Index (TLI), accounting for degrees of freedom, indicated a better fit. The standardized root mean square of residuals (SRMR), measuring the difference between observed and implied correlation matrices, was notably lower at 0.04, well below the recommended ceiling of 0.08. Lastly, the standardized root mean square error of approximation (RMSEA) value was quite low at 0.07, suggesting a moderate to acceptable fit of the data to the model (see Table 6).

Observing these results, the model fitness indices collectively indicate a good fit of the data. While the chi-square statistic did not reject the null hypothesis of a good fit, other indices such as NFI, IFI, CFI, TLI, SRMR, and RMSEA consistently support the conclusion of a well-fitted model. These indices consider factors such as size, sample characteristics, and degrees of freedom, providing a comprehensive evaluation of the overall fit of the model.

**Table 6. Measures of fit indices of latent constructs in perceived access to health-care services among patients with cervical cancer visiting oncology centers in Addis Ababa, Ethiopia.**

| Fit measures | Fit indices of the default model |
| --- | --- |
| CMIN | 386 |
| CMIN/df | 2.82 |
| IFI | 0.95 |
| NFI | 0.93 |
| CFI | 0.95 |
| RFI | 0.91 |
| TLI | 0.94 |
| SRMR | 0.04 |
| RMSEA | 0.07 |

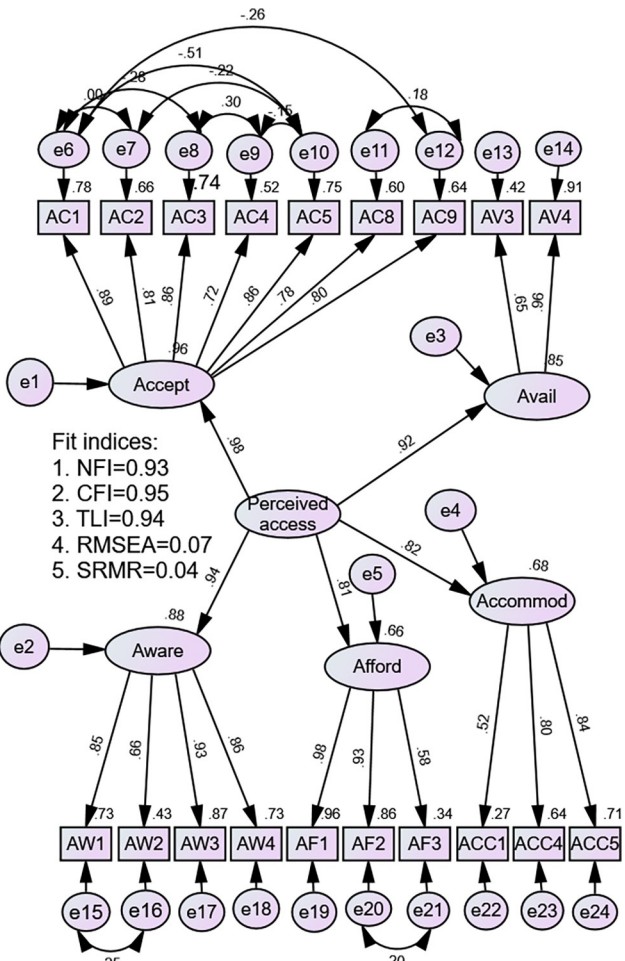

**Fig 1. Structural equation modeling of the construct structure in perceived access to healthcare services among patients with cervical cancer visiting oncology centers in Addis Ababa, Ethiopia.**

The second-order structure of the instrument, comprising 19 items and 5 latent domains, along with selected model fit indices, is illustrated in the figure below (Fig 1).

## Discussion

The objective of this study was to validate the Amharic version of a psychometric measure assessing the perceived access to health-care services among patients with cervical cancer in Ethiopia. The research adhered to the necessary epistemic procedures essential for a psychometric test. Significantly, it addressed a crucial aspect of the validation process by not only focusing on the measures of interest but also ensuring the validity of the employed procedures [59]. Validating the measure in the Ethiopian context is crucial as it ensures that the instrument is culturally and linguistically appropriate, contributing to the accurate assessment of perceived access to healthcare services among the target population in Ethiopia. This validation enhances the reliability and applicability of the measure in the local context, facilitating more meaningful and culturally sensitive research in the field of cervical cancer care in Ethiopia. Furthermore, preparing and validating a local-language version of an instrument is deemed beneficial to ensure stability of measures it stands for and eases comparison of

findings across similar settings. The requirement for a valid tool in such researches stems from the need in a theoretically founded and precise comprehension of abstract concepts commonly encountered in health sciences [60].

The initial activity (phase I) involved the design and adaptation of the original instrument, encompassing both qualitative and quantitative tasks. This was succeeded by piloting the instrument (phase II) and conducting factorial analysis (phase III), with the latter two phases discussed in subsequent paragraphs. During the qualitative evaluation, domain-specific items were revised so that the scope and relevance of an item would be examined to fit in the present context. As such, consideration from the earlier sources [1, 4, 8] were taken into account with an effort to fit the population of interest, treatment setting and relevance of a variable. This is an important element in cross-cultural adaptation of an instrument since failure to consider current settings may result in deviated outputs [61]. Almanasreh et al., (2019) too regarded content validation a necessary step as an item is identified than quantitatively scored [40]. In other words, the expert panels were to evaluate and decide which items to be included in the new instrument, how the wording should be put in the Amharic version and if some words should be replaced with synonyms (for example, screening for colon or breast cancer instead of cancer). In the quantitative component, numeric ratings were assigned to the items within each domain. These scores were then aggregated and compared against established thresholds. As a result, the outcomes from the subsequent calculations fell within the acceptable range, indicating a consensus among expert ratings. The minimum value in the CVR, I-CVI, and S-CVI were 0.5, 0.75, and 0.91 respectively. Out of a total of 30 items, only two items scored an I-CVI of 0.75 whereas, in the remaining, it was 0.88 or higher. Hence, most of the scores are in agreement with suggested ranges in the literature [8, 62]. More specifically, quite conservative reports in the literature have been reported for an excellent I-CVI score falling at 0.78 or higher [36, 44, 63].

After the successful assessment of the initial phase and the reliability evaluation of the piloted instrument on a limited sample size (Phase II), it was subsequently employed on a larger sample of cases for the assessment of factorial validity. The internal consistency of items in the factor analysis was found to satisfy the model excellently for most of the constructs. Sequential improvement was sought for some of the dimensions by dropping items from the construct. In fact, this was carried out only for those unidimensionality was severely disturbed and not in cases where the minimum level of at least 0.7 was attained. There is evidence in the literature claiming that a very high alpha value is not always trustable as it may offer limited reliability, especially, when testing a scientific knowledge [64]. In addition, we restricted item inclusion with a factor loading 0.4 or higher both for the first and second order model structures. This follows the recommendation proposed by Stevens (2012) as the minimum cut-off criteria for factor loading in structural equation modeling [65]. On the other hand, Hair et al., (2011) argued that a standardized factor loadings of at least 0.5 for an item or a construct (for higher orders) to explain 25% of the variance of a construct of interest [66]. Accordingly, the final model, in the present study, contained five domains with 19 observed items. One of the first order constructs 'reachability' was not included in the matrix structure of the second order CFA as its loading value was extremely low (0.12). Alternatively, the six-domain model could also be structured with acceptable fit assuming independent identification of factors. However, perceived access to healthcare services should better be conceptualized as second order construct from the other first order constructs (accessibility, acceptability, availability, accommodability, and awareness) as it offers a reasonable theoretical basis of the usual healthcare care practice. The observed high internal consistency indicates strong coherence among items within constructs. The decision to selectively drop certain items, guided by factor loading criteria, enhances the instrument's applicability and reliability. By retaining only items

with substantial contributions, the refined instrument becomes more precise, aligned with theoretical foundations, and offers improved reliability and validity in assessing perceived access to healthcare services.

Other measures of validity namely, convergent validity, divergent validity and known-group validity were performed. The result of convergent validity of items per each construct was such that both composite reliability (CR) and average variance extracted (AVE) met the acceptable value of at least 0.6 and 0.5 respectively as suggested elsewhere [67, 68]. Divergent validity assessment using the improved HTMT2 methods [54] revealed a good fit to the threshold except a couple of the constructs suggesting presence of multicollinearity, i.e the factors acceptability and availability and acceptability and awareness showed a concern of discriminant validity. As opposed to items meant for measuring poorly related concepts in which case the value of discriminant validity would expected to be low [69], this result could be viewed as to the fact that the constructs considered are substantially correlated. The observed multicollinearity between certain constructs suggests potential overlap or correlation between them. This could arise due to shared conceptual elements or interconnected aspects of the constructs being measured. The presence of multicollinearity may impact the interpretation of results by making it challenging to isolate the unique contributions of each construct. In other words, the high correlation between certain factors may lead to challenges in attributing variations in the outcome specifically to one construct, potentially complicating the understanding of the individual effects of those correlated constructs. The significant multicollinearity observed between these pairs of constructs may suggest their interconnected nature rather than indicating a flaw that needs further refinement, particularly in the case of acceptability vs. awareness. However, a thoughtful examination and exploration are warranted to better understand the interconnectedness of acceptability vs. availability in the healthcare context. Likewise, the instrument was able to discriminate between independent groups of the sample with considerable discrepancy of the outcome measure. It is presumably sound to accept that people at least with different income level [70], status of community-based health insurance coverage [71], presence of social support [72] and employment status [6] do act or react to access of healthcare services quite differently. The convergent validity findings assure that the instrument accurately measures intended constructs, enhancing its quality. Divergent validity identifies potential correlations, providing nuanced insights. Known-group validity signifies the instrument's ability to discriminate among diverse demographic groups, enhancing its utility across populations. Collectively, these findings affirm the instrument's quality, reliability, and applicability in assessing perceptions of healthcare access.

The model also demonstrated that the commonly utilized incremental fitness indices, namely CFI and TLI were achieved with the corresponding values of 0.95, and 0.94. This suggests that the sample data was able to fit the model adequately compared to the baseline (worst case) model. On the other hand, absolute fit measures, namely RMSEA and SRMAR were still in the acceptable range with corresponding values of 0.07 and 0.04 respectively. These indices measure how the hypothesized data fits the model compared to the perfect model. Even though it has been reported that RMSEA should be 0.05 or lower in order for a model achieves excellent fit [73], Chen et al. also argued that this classification does not have an empirical support to identify an absolutely good fit [74]. They proposed that other factors, such as degrees of freedom and sample size should also be taken into account. A level at or below 0.08 has also been suggested acceptable fit [36]. The SRMR is a measure of badness of model fit measured from the residuals covariance and the implied sample with values closer to zero indicating a characteristic of good fit [75]. The incremental fitness indices, CFI and TLI, indicate that the sample data fits the model adequately compared to the baseline model, enhancing confidence in the instrument's validation. The acceptable range of absolute fit measures, RMSEA (0.07)

and SRMR (0.04), suggests reasonable model fit. While the commonly cited threshold for excellent fit is an RMSEA of 0.05 or lower, it's important to consider additional factors like degrees of freedom and sample size. The SRMR, with values closer to zero, further supports the instrument's good fit. Overall, these nuanced interpretations collectively contribute to the comprehensive validation of the instrument.

The current study aimed to refine and validate the Amharic version of perceived access to healthcare services in the Ethiopian context, where no similar method was available. The model utilized a hierarchical second-order analysis, aligning with the theoretical conception of the phenomenon under investigation. Results obtained from the sample were compared with bootstrapped output, incorporating 95% confidence intervals to mitigate the impact of skewed distribution. Additionally, the study adhered to a series of recommended steps during the reliability and validity testing processes. However, limitations included an inadequate sample size impacting generalizability, a lack of discrimination between constructs affecting precision. In addition, relying on constructs from previous literature may introduce bias and limit adaptability to the specific Ethiopian context, potentially influencing outcomes by overlooking crucial factors.

## Conclusion

The Amharic version of perceived access to health-care services revealed a good fit to be utilized by Ethiopian patients with chronic illness, particularly of cancer diagnosis. The developed scale embraces multidimensional psychometric domains which comprising of acceptability, availability, accommodability, affordability and awareness from patients perspective. The introduction into practice of this instrument will be useful for researchers, health service providers and policy makers to easily capture the perception of patients on their behavior towards demanding and consuming health. This will then guide their practice of service delivery and set for appropriate intervention plans. While the outcomes obtained through the method will be prone to reports of subjective nature, we recommend that researches and quality improvement initiatives that assess access to health-care utilization of patients also accompany with objective methods of measurement. We also suggest that the tool can be utilized in other patient populations with a consideration of additional constructs, such as geographic accessibility. Finally, future studies should prioritize larger samples, refine measurement items, and adopt a more context-specific approach to construct definition.

## Supporting information

**S1 File.**
(DOCX)

**S2 File.**
(DOCX)

**S3 File.**
(DOCX)

## Acknowledgments

The authors express gratitude to all patients with cervical cancer who willingly participated in this study. Additionally, we acknowledge the kind support and facilitation received from the respective hospital officials during the data collection process.

## Author Contributions

**Conceptualization:** Tariku Shimels, Teferi Gedif Fenta.

**Data curation:** Tariku Shimels.

**Formal analysis:** Tariku Shimels.

**Funding acquisition:** Tariku Shimels.

**Methodology:** Tariku Shimels.

**Project administration:** Tariku Shimels.

**Resources:** Teferi Gedif Fenta.

**Supervision:** Tariku Shimels, Biruck Gashawbeza, Teferi Gedif Fenta.

**Validation:** Tariku Shimels, Biruck Gashawbeza, Teferi Gedif Fenta.

**Visualization:** Tariku Shimels.

**Writing – original draft:** Tariku Shimels.

**Writing – review & editing:** Tariku Shimels, Biruck Gashawbeza, Teferi Gedif Fenta.

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
