## [Decision Letter · Decision Letter 0]

28 Dec 2023

PONE-D-23-34498Validating the Amharic version of perceived access to healthcare services in patients with cervical cancer in Ethiopia: A second-order confirmatory factor analysisPLOS ONE

Dear Dr. Shimels,

Thank you for submitting your manuscript to PLOS ONE. After careful consideration, we feel that it has merit but does not fully meet PLOS ONE’s publication criteria as it currently stands. Therefore, we invite you to submit a revised version of the manuscript that addresses the points raised during the review process.

We look forward to receiving your revised manuscript.

Kind regards,

Roghieh Nooripour, Ph.D

Academic Editor

PLOS ONE

Journal Requirements:

3. Please ensure that you refer to Figure 1 in your text as, if accepted, production will need this reference to link the reader to the figure.

Reviewers' comments:

Reviewer's Responses to Questions

**Comments to the Author**

1. Is the manuscript technically sound, and do the data support the conclusions?

Reviewer #1: Partly

Reviewer #2: Yes

2. Has the statistical analysis been performed appropriately and rigorously? 

Reviewer #1: N/A

Reviewer #2: Yes

3. Have the authors made all data underlying the findings in their manuscript fully available?

Reviewer #1: Yes

Reviewer #2: Yes

4. Is the manuscript presented in an intelligible fashion and written in standard English?

Reviewer #1: Yes

Reviewer #2: No

5. Review Comments to the Author

Reviewer #1: The topic is fascinating, but a thorough restructuring of the content is necessary. I kindly request the esteemed author to revise the entire article with a deeper and more coherent perspective, taking into consideration the suggested changes. Afterward, please resubmit it for further review. I believe this feedback will be valuable in improving the quality of your work.

Best regards.

Abstract

1. Be more specific about the objectives of your study. Clearly state how validating the Amharic-version of the instrument contributes to the field.

1. Provide more details on the consecutive sampling method. Clarify how it ensures a representative sample of the population.

1. Provide more in-depth interpretation of your results. Discuss the implications of the internal consistency and validity tests.

Introduction

• Clarify and distinguish between the various definitions and dimensions of access to healthcare. Consider using a table or a diagram to visually represent the different models and dimensions proposed by various authors.

• Provide more context or examples to illustrate the differences between the dimensions of access, such as "affordability," "availability," "accessibility," etc.

• Strengthen the link between the various theories and the context of your study. Explicitly mention how these theories are relevant to the Ethiopian context.

• Include a brief discussion on how the understanding of access to healthcare has evolved over time, and what gaps your study aims to fill.

• Emphasize the relevance of the study to the Ethiopian context early in the section. Discuss the specific challenges related to healthcare access in Ethiopia and how they relate to the models and dimensions of access you've described.

• Provide statistics or recent studies highlighting the healthcare access situation in Ethiopia, especially in relation to cervical cancer care.

• Discuss the limitations of existing methods to measure patient perceptions of healthcare access. Highlight why these may be inadequate or less applicable in the Ethiopian context, paving the way for the need for your study.

• Mention any specific shortcomings in previous studies that used these methods in similar contexts, and how your study addresses these shortcomings.

• Ensure the section flows logically from the general understanding of healthcare access to the specifics of the Ethiopian context.

• Avoid jargon or overly technical language to make the section accessible to readers from various backgrounds.

Method

1. Expand the description of the oncology centers in Addis Ababa to provide context for readers unfamiliar with the Ethiopian healthcare system. This might include the capacity of these centers, typical patient demographics, etc.

2. Explain why a cross-sectional design is appropriate for this study and how it helps in achieving the study objectives.

3. Provide more detailed reasoning for choosing the specific validity and reliability tests used. This includes why these tests are appropriate for your study population and the instrument being validated.

4. Discuss any potential limitations of these methods and how they might affect the interpretation of your results.

5. Clarify how the psychometric domains and constructs were specifically adapted from Hoseini et al.'s questionnaire to fit the Ethiopian context and the specific needs of cervical cancer patients.

6. Explain the rationale behind each modification made to the original instrument.

7. Detail the qualifications and expertise of the translators to ensure their competency in translating a psychometric instrument.

8. Discuss how you addressed any cultural nuances in translation that might affect the interpretation of the questions.

9. Elaborate on how the panel of experts was selected, including their expertise and relevance to the study.

10. Explain how the feedback from the panel was incorporated into the final instrument.

11. Describe the characteristics of the pilot sample and how they compare to the main study population.

12. Discuss any revisions made to the instrument following the pilot study and the rationale behind these changes.

13. Justify the use of Cronbach's alpha and confirmatory factor analysis in the context of your study.

14. Discuss how these analyses contribute to the overall validation of the instrument.

15. Explain the training provided to data collectors and how it contributes to the quality and consistency of the data collected.

16. Describe any quality control measures taken during data collection.

17. Elaborate on the choice of statistical methods and software used for data analysis.

18. Discuss how these methods are suitable for the type of data collected and the research questions posed.

19. Explain the process of obtaining informed consent and how the study ensures confidentiality and privacy of the participants.

Results

• Provide a brief interpretation of the participant characteristics, highlighting any notable findings such as the high number of referred patients from outside Addis Ababa or the demographic trends observed.

• Consider discussing the implications of these characteristics on the generalizability of the study findings.

• Elaborate on the rationale for modifying specific components of the instrument, such as changing 'access' to 'reachability' and the inclusion of different cancer types.

• Discuss the implications of the content validity ratio (CVR) and item-level content validity index (I-CVI) values obtained. Explain what these values suggest about the quality of your instrument.

• Provide an interpretation of the Cronbach’s alpha values obtained for each factor. Discuss what these values indicate about the reliability of the instrument.

• Clarify the decision-making process behind the removal of specific items from the factors based on the reliability analysis.

• Interpret the composite reliability (CR) and average variance extracted (AVE) for each factor. Explain how these values confirm the convergent validity of the instrument.

• Discuss any factors that did not meet the recommended thresholds and what this might imply.

• Explain the significance of the hetero-trait to mono-trait (HTMT2) ratio results. Discuss the implications of the two factor pairs that showed higher values than the recommended threshold.

• Interpret the findings from the known group validity analysis. Discuss how the instrument differentiates between groups based on income, health insurance status, social support, and occupation.

• Explain the significance of these findings in the context of assessing perceived access to healthcare.

• Provide a detailed interpretation of the model fitness indices. Discuss what each index (e.g., CFI, TLI, SRMR, RMSEA) indicates about the overall fit of your model.

Discussion

o Reiterate the study's primary aim at the beginning of the discussion to remind readers of the context. Link back to how your findings fulfill this aim, emphasizing the significance of validating the Amharic version of the psychometric measure in the Ethiopian context.

o Discuss the implications of the high internal consistency and the decision to drop certain items in more detail. Explore how these adjustments improve the instrument's applicability and reliability.

o Explain the significance of the convergent, divergent, and known-group validity findings in relation to the quality and utility of the instrument.

o Delve deeper into the multicollinearity observed between certain constructs. Discuss potential reasons for this overlap and how it might affect the interpretation of results.

o Consider whether this finding suggests a need for further refinement of the instrument or reflects the interconnected nature of these constructs in the context of healthcare access.

o Provide a more nuanced interpretation of the model fitness indices, discussing how each index contributes to the overall validation of the instrument.

o Address the moderate fit indicated by RMSEA and what it might imply about the model's applicability.

o Compare and contrast your findings with previous studies that have attempted similar validations in different settings or populations. This will help readers understand the unique contributions of your study.

o Elaborate on each limitation mentioned. For instance, discuss how the sample size might have impacted the findings and what could be done in future studies to mitigate this.

o Address the potential impact of relying on constructs from previous literature and how this might have influenced the outcomes.

Reviewer #2: I would thank the Authors to have the possibility to read this work.

I will provide some suggestions to improve the manuscript.

Firstly, the paper must be intesively revised considering the English language and the grammatical in some parts.

Then, it could be better to explain the reason why, for the study, the Authors decided to use the questionnaire provided in the literature by Hoseini and colleagues.

The methodology results well structured and explained even if, in some parts, it could be better to describe in a more appropriate way the experts' panel involved, also dividing the methodological aspects and the used materials from the obtained results and the calculated statistical indexes.

The last comment is related to the anonimity and the ethical concerns of the study. In the methodology it is defined that the participants provided a verbal informed consent, but a written informed consent was not provided?

I think that improving these aspects the manuscript may be ready for the publication!

All the best!

6. PLOS authors have the option to publish the peer review history of their article (what does this mean?). If published, this will include your full peer review and any attached files.

Reviewer #1: No

Reviewer #2: No

---

## [Author Response · Author response to Decision Letter 0]

15 Jan 2024

Dear reviewers,

Kindly, find the point-by-point responses provided in the separate attachment designated 'response to reviewers' comments'. 

We appreciate for your time and consideration on this particular submission.

With kindest regards,

The corresponding author

---

## [Editor Report · Decision Letter 1]

1 Feb 2024

PONE-D-23-34498R1

Validation of the Amharic Version of Perceived Access to Healthcare Services for Patients with Cervical Cancer in Ethiopia: A Second-Order Confirmatory Factor Analysis

PLOS ONE

Dear Dr. Shimels,

Thank you for submitting your manuscript to PLOS ONE. After careful consideration, we feel that it has merit but does not fully meet PLOS ONE’s publication criteria as it currently stands. Therefore, we invite you to submit a revised version of the manuscript that addresses the points raised during the review process.

We look forward to receiving your revised manuscript.

Kind regards,

Academic Editor

PLOS ONE

**Academic Editor comments**
**:**

Introduction

Provide a bit more context regarding the Ethiopian healthcare system and any unique challenges it faces. This will help readers unfamiliar with the Ethiopian context to better understand the significance of the study.To enhance the quality of your introduction, consider incorporating the following referenceshttps://link.springer.com/article/10.1007/s12144-021-01662-2https://pubmed.ncbi.nlm.nih.gov/35943005/https://www.archbreastcancer.com/index.php/abc/article/view/470
Ensure that terminology is consistently defined. For instance, the term "psychometric component" could be briefly explained for readers who may not be familiar with it.

Discussion

Provide more detailed explanations of the steps involved in the validation process. For instance, elaborate on the qualitative and quantitative approaches used for content validity and explain the significance of each step.Emphasize the importance of future research efforts focusing on larger sample sizes, refinement of measurement items, and the adoption of a more context-specific approach to construct definition.

---

## [Author Response · Author response to Decision Letter 1]

5 Feb 2024

Dear Esteemed Editor,

We extend our sincere gratitude to you for the invaluable comments and suggestions you provided. Your insights have been truly enlightening and have significantly enhanced the quality of the manuscript. Your expertise and attention to detail have made a substantial impact on the clarity and coherence of the content. We appreciate the time and effort you invested in reviewing the manuscript thoroughly. Your constructive feedback has not only improved the overall structure but has also contributed to a more comprehensive and robust presentation of the research findings.

Once again, thank you for your dedication to ensuring the excellence of this work. We hope that we have addressed all your comments and suggestions. Please let us know if you have more enquiries.

Kindest regards,

Tariku S, corresponding author 

S.no Editor comments Responses 

1 Provide a bit more context regarding the Ethiopian healthcare system and any unique challenges it faces. This will help readers unfamiliar with the Ethiopian context to better understand the significance of the study. Thank you for the insightful comment. We have included a further detail about the Ethiopian health system, the national plan on cancer control and the status of cancer, especially, cervical cancer despite the efforts. Kindly find the edits on page 5. 

2 To enhance the quality of your introduction, consider incorporating the following references

1. https://link.springer.com/article/10.1007/s12144-021-01662-2

2. https://pubmed.ncbi.nlm.nih.gov/35943005/

3. https://www.archbreastcancer.com/index.php/abc/article/view/470

 Thank you for the suggestion. We have now included the recommended sources in the introduction section. Please see the highlighted text on page 4. 

3 Ensure that terminology is consistently defined. For instance, the term "psychometric component" could be briefly explained for readers who may not be familiar with it.

 Thank you for the comment. An extended explanation has been added following your concern. Kindly find the added edits at the end of the introduction on page 5. 

4 Provide more detailed explanations of the steps involved in the validation process. For instance, elaborate on the qualitative and quantitative approaches used for content validity and explain the significance of each step Thank you for the comment. We have stated the main activities performed in the validation process as design and adaptation (phase I), pilot testing (phase II), and factorial analysis (phase III) at the beginning of paragraph 2 in the discussion. We have then extended the discussion on the process and significance of the qualitative and quantitative methods used. Please see highlighted text on page 21. 

5 Emphasize the importance of future research efforts focusing on larger sample sizes, refinement of measurement items, and the adoption of a more context-specific approach to construct definition.

 Thank you for the concern and suggestion. While the limitations of the study raised those issues at the end of the discussion section, a recommendation on the same has been added at the end of the conclusion section. Kindly check the highlighted text on pages 21 &22.

---

## [Editor Report · Decision Letter 2]

6 Mar 2024

Validation of the Amharic Version of Perceived Access to Healthcare Services for Patients with Cervical Cancer in Ethiopia: A Second-Order Confirmatory Factor Analysis

PONE-D-23-34498R2

Dear Dr. Shimels,

We’re pleased to inform you that your manuscript has been judged scientifically suitable for publication and will be formally accepted for publication once it meets all outstanding technical requirements.

Kind regards,

Roghieh Nooripour, Ph.D

Academic Editor

PLOS ONE

---

## [Editor Report · Acceptance letter]

10 Mar 2024

PONE-D-23-34498R2 

PLOS ONE

Dear Dr. Shimels, 

I'm pleased to inform you that your manuscript has been deemed suitable for publication in PLOS ONE. Congratulations! Your manuscript is now being handed over to our production team.

Kind regards, 

on behalf of

Dr. Roghieh Nooripour 

Academic Editor

PLOS ONE